# Association between Fathers’ and Mothers’ Parenting Styles and the Risk of Overweight/Obesity among Adolescents in San José Province, Costa Rica

**DOI:** 10.3390/nu14245328

**Published:** 2022-12-15

**Authors:** Rafael Monge-Rojas, Vanessa Smith-Castro, Teresia M. O’Connor, Rulamán Vargas-Quesada, Benjamín Reyes-Fernández

**Affiliations:** 1Nutrition and Health Unit, Costa Rican Institute for Research and Education on Nutrition and Health (INCIENSA), Ministry of Health, Tres Ríos P.O. Box 4-2250, Costa Rica; 2Psychological Research Institute, Universidad de Costa Rica, City of Research, Montes de Oca 11501-2060, Costa Rica; 3USDA/ARS Children’s Nutrition Research Center, Baylor College of Medicine, Houston, TX 77030, USA

**Keywords:** parenting styles, overweight, obesity, Costa Rica, adolescents

## Abstract

Parenting styles are a risk factor for adolescents overweight/obesity worldwide, but this association is not well understood in the context of Latin America. This study examines the association between the parenting styles of mothers and fathers and the risk of overweight/obesity among Costa Rican adolescents. Data are cross-sectional from a sample of adolescents (13–18 years old) enrolled in ten urban and eight rural schools (*n* = 18) in the province of San José, Costa Rica, in 2017. Hierarchical logistic regression analyses were performed to assess the likelihood of adolescents being overweight according to the mothers’ and fathers’ parenting styles. A significant association was found between the risk of adolescent overweight/obesity and the paternal authoritarian style only in rural areas (*B* = 0.622, *SE* = 0.317, *Wald* = 3.864, *ExpB* = 1.863, *p* = 0.04), and between said risk and the paternal permissive style only in male adolescents (*B* = 0.901, *SE* = 0.435, *Wald* = 4.286, *ExpB* = 2.461, *p* = 0.038). For maternal parenting styles, no associations reached significant levels once logistic regression models were adjusted for the fathers’ parenting styles. These findings underscore the importance of further studying the role of fathers’ paternal parenting styles on Latin American adolescent weight outcomes. Expanding our understanding of the parenting styles of fathers has important implications for the design and implementation of culturally- and gender-appropriate family interventions.

## 1. Introduction

Obesity during adolescence is associated with an increased risk of cardiovascular disease, cardiometabolic syndrome, and related complications into adulthood [1]. The Latin American region has been driving large increases in the global prevalence of adolescent obesity [2,3]. Specifically, in Costa Rica, recent data showed that 21% and 9.8% of adolescents (13–18 y) suffer from overweight and obesity, respectively [4]. This finding is quite alarming because, out of five Central American countries, Costa Rica already has the lowest percentage (9%) of adults (20 years or older) that are free of any metabolic syndrome components [5].

Although it has been considered that the fundamental cause of overweight/obesity is an energy imbalance between calories consumed and calories expended, several psychosocial factors contribute to the development and maintenance of overweight and obesity during adolescence [6,7]. There is strong evidence for the influence of the family environment and the social context on weight-related outcomes in this developmental age period [6,8,9].

Specifically, parenting styles have been associated with weight status in school-aged children and adolescents [9,10,11,12,13]. Parenting styles aim to define how parents interact with children as they grow [13]. The styles are a constellation of attitudes toward the child that, taken together, create an emotional climate in which parental behaviors are expressed [14].

Some studies have shown that parenting styles that are authoritarian (low responsiveness, high demandingness) and permissive/indulgent (low responsiveness, low demandingness) are related to higher BMIs in the offspring [13,15,16,17,18,19], whereas an authoritative parenting style (high responsiveness, high demandingness) are related to a lower BMI in the child [10,11,15,17,20]. However, since most of these studies use the term “children” to refer to the 3-to-16-year-old group, less is known about the association between parenting styles and overweight risk among adolescents aged 13–18.

Published studies on adolescents have been primarily conducted in the United States and on maternal (rather than paternal) parenting styles. In these studies [9,12,13], having either authoritarian or disengaged parents is associated with larger BMI increases as the children transitioned to young adulthood, regardless of their previous BMI trajectory [9]. The maternal authoritarian style has been identified as a risk factor for higher BMIs among male adolescents [13], whereas the authoritative style in mothers could play a protective role against the development of overweight among adolescents of both sexes [12,13]. These findings suggest that maternal parenting styles can have long-lasting effects on a child’s weight regulation throughout their life. Less is known about the impact of paternal parenting styles or the combined effects of both maternal and paternal styles on the weight status of adolescents. Fathers are often underrepresented in observational studies on parenting styles and obesity studies in general [21]. The limited data on fathers has not found a significant association between paternal parenting styles and adolescent BMIs [13,21]. Further, if parenting styles between the mothers and fathers differ considerably, these incongruent styles may exert diverse influences on adolescent BMIs. A study in the United States suggests that the co-occurrence of incongruent parenting styles (authoritarian mother and neglectful father) can increase the likelihood of higher BMIs among adolescent males [13]. According to Berge and colleagues [13], incongruent parenting styles can foster a chaotic or inconsistent home environment that can affect the adolescent’s regulation of eating or physical activity.

Parenting styles are influenced by diverse sociocultural factors. Differences in the fundamental basics of parenting styles are led by discrepancies in thoughts and behaviors among cultures, and their results can vary between urban and rural communities in the same country, as well as according to the sex of the child. For example, studies in the Middle East and South Asian countries suggest that the authoritarian style is more prevalent in urban areas, while the authoritative style predominates in rural areas [22,23,24,25]. Parenting styles may depend on the child’s sex, according to the cultural context in which the parent–child dyad socializes, even within the same urban or rural environment [22,23,24,25].

Individualistic cultures, common in North American and European countries, support emotional independence, assertiveness, autonomy, and the need for privacy where the individual loosens their bonds with others [26]. On the contrary, collectivist cultures, which prevail in Costa Rica and other Latin American countries [27], prioritize socialization, obedience, security, and family integrity [26]. The differences between the two cultures lead to dissimilar parenting styles because parenting behaviors and their impacts vary from one culture to another [28]. Therefore, characterizing the association between the fathers’ and mothers’ parenting styles and the risk of being overweight among adolescents in a Latin American context is an essential first step to contribute to the design of culturally appropriate interventions that promote healthy weights in this demographic group.

The objective of this study is to describe the association between paternal and maternal parenting styles and adolescent overweight and obesity risk by the sex of the child and by their area of residence in Costa Rica. In this way, the study aims to significantly expand the existing literature on parenting styles and adolescent weight in three key ways: (1) purposively including the fathers’ parenting styles in addition to the mothers’; (2) examining the associations by the sex of the adolescent offspring; and (3) in the context of Latin America.

## 2. Materials and Methods

### 2.1. Study Population and Setting

The study population was drawn from Costa Rican adolescents (13–18 years old, 7th to 11th graders) enrolled in rural and urban schools in the province of San José. This province was selected because it has the highest concentration of adolescents (30%) in the country [29], and the majority (80%) of them are enrolled in school [30]. Adolescents were recruited between February 2017 and September 2017.

The sample size was determined assuming a sampling error for a population proportion with a finite population correction [31]. Urban and rural schools were selected using a proportional-size probability method [32]. In each school, ten classrooms (two from each grade from 7th to 11th) were selected using simple random sampling. Students from each classroom were invited to participate in the study and provided informed assent and informed consent forms. In order to define the study sample, a random selection was made among those students who provided signed informed consent and assent forms. About 5% of the selected students decided not to participate in the study before the start. The final study sample was 818 adolescents aged 13 to 18 years. Only those who provided complete data on both their fathers’ and mothers’ parenting styles were included in this analysis (*n* = 695).

### 2.2. Data Collection

At each high school, the adolescents were gathered in a classroom reserved for the study during regular hours. A researcher instructed the students on completing their sociodemographic information (age, gender, and area of residence), and filling out the Parenting Styles Questionnaire. One of the researchers involved in this study was available to answer any questions. The adolescents were weighed and measured upon completion of the scale as described further below.

### 2.3. Parenting Styles Questionnaire

To assess their perception of their parents’ parenting styles, adolescents filled out the short version (32 items) of the Parenting Styles and Dimensions Questionnaire (PSDQ) [33]. This tool was designed to measure parenting styles according to Baumrind’s typologies: authoritative (high responsiveness and high demandingness), authoritarian (low responsiveness and high demandingness), and permissive/indulgent (high responsiveness and low demandingness) [34]. The Authoritative scale contains 15 items reflecting the three dimensions of warmth and support, regulation, and autonomy granting; the Authoritarian scale involves 12 items, yielding the three dimensions of physical coercion, verbal hostility, and nonreasoning/punitive strategies, and the Permissive scale comprises only one dimension, indulgence, which is composed of five items [33]. Items use a 5-point Likert scale ranging from Never (1) to Always (5). The score for each of the dimensions is the average of its items.

The PSDQ was translated into Spanish by the authors (native Spanish speakers from Costa Rica) since, at the time of data collection, there was no published literature on the short version of the questionnaire for Spanish-speaking audiences. Further, 20 adolescents (13–18 y) in Costa Rica were polled using cognitive interviewing techniques [35] to evaluate survey item comprehension, and the questions were later revised to increase understanding.

Our team performed a psychometric validation of the translated PSDQ short version with the Costa Rican adolescent study population having both mothers and fathers (*n* = 300). Therefore, participating adolescents were asked to complete the PSDQ twice to assess their perception of the warmth and demandingness of both their fathers and mothers. Both parents were scored on each of the three parenting styles. This validation has been published elsewhere [36], but in brief, the paternal and maternal authoritative and authoritarian parenting styles scales had an internal consistency that ranged between satisfactory and good [37] (paternal: Cronbach α = 0.91 and 0.70, respectively; maternal: Cronbach α = 0.90 and 0.73, respectively). However, the permissive parenting style scale had a moderate consistency for mothers (Cronbach α = 0.51) and fathers (Cronbach α = 0.52).

Although the PSDQ was designed to be used with parents of children aged 4 to 12, the questionnaire has been validated and used in several adolescent studies to investigate various health outcomes [38,39,40,41,42].

### 2.4. Anthropometric Assessment

A trained nutritionist measured participants’ heights and weights. Height was obtained to the nearest 0.1 cm using a stadiometer, and weight was measured in kg using a digital scale following the methodology described by Preedy [43]. Before stepping on the digital scale, adolescents were asked to remove their shoes and as much outerwear as possible and to empty their pockets. Body Mass Index (BMI) values were calculated from measured heights and weights using the standard equation: weight (kg)/height (m)^2^. Nutritional status was determined using the BMI Z score for age, as recommended by the World Health Organization [44]: <−2: underweight; ≥−2 and <+1: healthy weight (eutrophy); ≥+1 and <+2: overweight, and ≥+2: obese. The outcome for all assessments was overweight/obese with a BMI Z score of ≥+1.

### 2.5. Sociodemographic Variables 

Data on sex, age, area of residence, parental education level, ownership of goods, and access to services (e.g., computers, internet, router, cable television, and water heating for the whole house), and family structure were collected using a paper-based questionnaire. The information on educational level and ownership of goods and access to services was used to determine the adolescents’ socioeconomic status according to the socioeconomic status index for students and secondary schools in Costa Rica developed by Madrigal and Gómez [45] and using the k-means procedure [46].

### 2.6. Data Analysis

Two mixed ANOVAs for paternal and maternal parenting styles were conducted to test for differences across the perceived parenting styles among healthy weight and overweight adolescents, with parenting styles as a within-subjects factor and weight as a between-subjects factor (healthy weight vs. overweight/obesity).

Various hierarchical logistic regression analyses [47] were performed to assess the likelihood of adolescents being overweight according to the mothers’ and fathers’ parenting styles. As there were some cases of obesity (*n* = 66), the combined overweight/obesity variable (henceforth overweight) was created to include adolescents with obesity (*n* = 66) and overweight (*n* = 161). This allowed for three-way interaction analyses while maintaining an adequate number of degrees of freedom.

In Step 1, maternal and paternal parenting styles, area of residence, SES, sex, and age were included as predictors of overweight. In Step 2, we included two-way interaction terms for each parenting style and adolescent sex and area of residence, respectively. In Step 3, we included three-way interactions for each parenting style, sex, and area of residence. All continuous variables were centered at their means, and binary predictors were dummy coded (0 = healthy weight, 1 = overweight/obese, 0 = boys, 1 = girls, 0 = urban, 1 = rural, 0 = low SES, 1 = middle/high SES). Statistically significant interactions were analyzed via a simple slope analysis [48]. The formula suggested by Paternoster et al. [49] was used to test the equality of regression coefficients. All statistical analyses were performed using the Statistical Package for Social Sciences (SPSS Inc., version 21.0 for Windows, Chicago, IL, USA). A *p*-value < 0.05 was considered statistically significant. The analytic plan was pre-specified, and any data-driven analyses were clearly identified and discussed appropriately.

## 3. Results

Table 1 presents the characteristics of 695 adolescents eligible for analysis: out of the total sample, 65% were girls, mean age was 14.9 years (SD 1.7), mean BMI was 22.3 (SD 4.3), and 50.2% lived in urban areas. The prevalence of overweight/obese in the sample was 33.5%. No differences were found across sex or area of residence.

Table 2 shows descriptive statistics for paternal and maternal parenting styles as perceived by adolescents. Although there were no significant differences across paternal and maternal parenting styles by adolescent weight status, the adolescents generally reported higher mean scores for authoritative styles, followed by permissive styles, and lower mean scores for authoritarian styles for both fathers and mothers (*F* _2.681_ = 370.84, *p* < 0.001 for fathers and *F* _2.687_ = 472.19, *p* < 0.001 for mothers). 

Table 3 summarizes the hierarchical logistic models predicting the influence of paternal and maternal parenting styles on adolescent overweight/obese status by area of residence and sex of the adolescent. For maternal parenting styles, no associations reached statistical significance once the fathers’ parenting styles were added to the models. However, paternal parenting styles were significantly associated with the risk of adolescent overweight/obesity in various manners. Analyses revealed a significant moderator effect by area of residence on the association between paternal authoritarian style and adolescent overweight/obesity (BMI Z score ≥ +1) (*B* = 1.403, *SE* = 0.66, *Wald* = 4.528, *Exp B* = 4.069, *p* = 0.033).

Simple slope analyses revealed a significant relationship between the paternal authoritarian style and the risk of overweight/obesity among adolescents living in rural areas (*B* = 0.622, *SE* = 0.317, *Wald* = 3.864, *ExpB* = 1.863, *p* = 0.04), but not in urban areas (*B* = −0.117, *SE* = 0.233, *Wald* = 0.254, *ExpB* = 0.889, *p* = 0.614). The odds of overweight/obesity among adolescents living in rural areas with authoritarian fathers were nearly twice those of urban adolescents (Figure 1). The test for equality of regression coefficients corroborated that the effect of authoritarian fathers on adolescent overweight/obesity varied significantly between urban and rural areas (*Z* = −1.878, *p* = 0.030).

Analyses also showed a significant three-way interaction between a permissive paternal style and adolescent sex and area of residence on overweight/obesity risk (*B* = 1.845, *SE* = 0.816, *Wald* = 5.113, *ExpB* = 6.328, *p* = 0.024). Two post hoc logistic regression analyses were conducted separately for each area of residence. In urban areas, a significant interaction was found between permissive fathers and adolescent sex on overweight/obesity risk (*B* = −1.297, *SE* = 0.538, *Wald* = 5.815, *ExpB* = 0.273, *p* = 0.016), but this interaction was not significant in rural areas (*B* = 0.465, *SE* = 0.619, *Wald* = 0.563, *ExpB* = 1.592, *p* = 0.453).

In urban areas, simple slope analyses revealed a significant relationship between a permissive paternal style and overweight risk only among male adolescents (*B* = 0.901, *SE* = 0.435, *Wald* = 4.286, *ExpB* = 2.461, *p* = 0.038) but not females (*B* = −0.409, *SE* = 0.323, *Wald* = 1.605, *ExpB* = 0.664, *p* = 0.205). Compared to females, males with permissive fathers had almost 2.5 times the odds of being overweight/obese (Figure 2). The test for equality of regression coefficients verified that the effect of permissive fathers on adolescent BMI risk varied significantly by adolescent sex (*Z* = 2.418, *p* = 0.007).

## 4. Discussion

Our results add significantly to the limited literature on the effects of paternal and maternal parenting styles on adolescent weight status. An important finding of our study is that only paternal parenting styles remained significantly associated with overweight/obesity risk among adolescents in Costa Rica, even when controlling for important predictors such as age and SES. This finding is not necessarily in agreement with most other studies, which have reported a significant association between maternal parenting styles and obesity risk in school-aged children and adolescents, although these have been conducted in the United States [10,11,12,20]. However, most other studies that have investigated the parenting styles of both parents and the children’s weight status are different from our study in that they combined maternal and paternal parenting styles as an overall ‘parenting style’, potentially skewing the results because overweight risk in adolescents may be influenced differently according to each parent’s parenting style [10]. When parenting styles are separated between mothers and fathers in heterosexual couples, the influence of the maternal parenting style may lose significance after adjusting for paternal styles, as happened in our analyses and four other prospective studies recently reviewed [10]. Another study that assessed the parenting style of fathers and mothers of 4–5-year-olds in Australia also demonstrated that the father’s, but not the mother’s, parenting style was associated with higher odds of their child having a higher weight status [50]. The reason why this occurs remains to be determined.

Our study includes a Latin American population that adds to the cultural diversity of the existing literature on parenting styles and adolescent overweight/obesity risk, which has previously focused mainly on non-Hispanic white populations. Therefore, the findings must be interpreted within the Hispanic and Latin American cultural context where values related to familism (e.g., loyalty, respect, obedience, and solidarity) may remain important, with a high tendency to prioritize hierarchy and respect for the elders, including parental figures [51]. Attitudinal familism in adolescents has been associated with the perception of parents serving as legitimate sources of guidance and unquestionable authority [52]. These attitudes have been historically present among Latin American parents compared to parents from other cultural backgrounds [53].

Nonetheless, cultures are constantly in flux, and Latin American adolescents, such as adolescents from other cultural contexts, are increasingly likely to believe that disagreeing with parents is acceptable and that behavioral autonomy is desirable [54]. This may cause conflict in the family, especially if authoritarian parents hold on to inflexible cultural values that emphasize unquestionable paternal authority, the heavy use of control and discipline, and a lack of warmth in the relationship.

Disagreements between parents and adolescents regarding the latter’s autonomy have been associated with alterations in the psychological adaptation of adolescents [53]. Especially among adolescents with authoritarian fathers, this parenting style may trigger emotional eating among adolescents and increase their risk of unhealthy weight gain [55]. Although attitudinal and behavioral familism has been identified as a protective factor for some adolescent outcomes, familism may also be potentially detrimental to adolescents living in stressful and conflictive contexts [54]. In rural areas of Costa Rica, where familial relational values are still deeply ingrained [56], our results suggest that authoritarian fathers may continue to pose a significant risk factor for adolescent overweight risk. New insights are needed to understand the complex dynamics occurring between fathers and adolescents, especially within the Latin American cultural context and as the adolescents’ sense of autonomy continues to increase.

Another factor at play in the interaction between fathers and adolescents is the sex of the child, as noted in our findings. Traditional Latino and Latin American cultures emphasize rights and responsibilities among family members based on sex [57]. However, boys are still raised in a machismo-endorsing society that grants them greater freedoms without strict parental monitoring [51]. Since boys are deemed more capable of looking after themselves, their social interactions may not be as closely monitored [58]. Role theory supports the notion that fathers may parent their sons and daughters differently [59], perhaps perpetuating the socialization roles accepted as masculine in the Latino culture. This context might help explain why fathers tend to favor a permissive parenting style towards boys.

Our findings show that a permissive parenting style is significantly associated with adolescent risk of overweight/obesity only in same-sex parent/adolescent dyads (father-son). This is in contrast to the findings Berge et al. [12] reported on opposite-sex parent/adolescent dyad patterns related to parenting styles and adolescent weight. Nonetheless, it is in agreement with previous evidence suggesting that Costa Rican adolescents have better communications with the same-sex parent [60] and that a permissive parenting style with the male adolescent [61] is more in line with the traditional machismo-endorsing society in the Latino culture [51]. It is critical to further study and expand our understanding of parenting styles and role theory within the Latin American context. Adolescent children of permissive parents tend to have less healthy home environments [62] and a higher risk of overeating tendencies [14,63], which may increase the risk of being overweight.

Considering the relevant role of parents in the risk of overweight/obesity evidenced in this study, researchers should look for strategies and facilitate parental participation in the studies. We must expand our understanding of the critical public health implications of paternal parenting styles, particularly for designing and implementing culturally and gender-appropriate family interventions aimed at preventing and reducing overweight among adolescents living in the San José province.

The underrepresentation of fathers in observational studies on parenting styles and the risk of overweight/obesity in the pediatric population has been associated with the traditional roles of guardians based on gender, lack of time, lack of interest in research, and lack of availability and accessibility compared to mothers [64,65]. However, a recent study with more than 300 parents showed that nearly 80% did not participate in research because they were not invited [65]. Davison [65] has suggested that parents are more likely to participate in studies when they perceive that the time they must allocate is short, the recruitment material emphasizes the benefits their participation would have for them and their families, and they receive information about the study through a parent-sensitive approach.

In conclusion, our results suggest that fathers of adolescent inhabitants of the province of San José have the potential to influence their children’s weight outcomes, perhaps even over and above the mothers’ influence. This shows that fathers’ and mothers’ parenting styles are related differently to the risk of overweight/obesity in adolescents living in this province. Understanding the influence of both paternal and maternal parenting styles on adolescent eating behaviors will help build up the knowledge of the emotional climates in which positive and negative nutritional behaviors originate.

This study had some strengths and limitations that must be considered when interpreting its results. The strengths include: (1) the study is the examination of urban and rural adolescents and their perceptions of both fathers’ and mothers’ parenting styles, as opposed to most studies that only include urban adolescents and maternal parenting styles; (2) by including fathers in our study, we enhanced opportunities to learn more about their influence on adolescent overweight- and obesity-related behaviors; and (3) the internal consistency analysis revealed mostly satisfactory to good values, with most scores exceeding the recommended minimum Cronbach’s alpha of 0.70 [37], except for the permissive scale which had moderate internal consistency (i.e., α = 0.52). This may have resulted from the limited number of items (*n* = 5) on the permissiveness scale as compared to 15 and 12 items on the authoritative and authoritarian scales, respectively. However, the moderate internal consistency of the permissiveness scale is similar to other studies [66] and the original questionnaire research [33]. Limitations include: (1) The study’s cross-sectional nature implies association, not causality, between parenting styles and adolescent overweight/obesity. Still, the study provides a perspective on the association between parenting styles and the risk of adolescent obesity within a Latin American cultural context, which is vastly different from the existing literature [9,12,13]. (2) Our study only involved adolescents and the results only reflect their perception of their parents’ parenting styles. Involving at least another family member would provide a more global perception of parenting styles. (3) The instrument we used did not categorize parents into one of the three parenting styles but instead provided scores across all three dimensions for each parent. Different results may have been found if parents were forced into one of the categories. (4) The study sample was not nationally representative; it was limited to urban and rural areas within the province of San José. However, the highest proportion of Costa Rican adolescents (30%) is clustered in that province [29]. Moreover, the sample included adolescents enrolled in school, representing ∼80% of all adolescents in Costa Rica [30].

## Figures and Tables

**Figure 1 nutrients-14-05328-f001:**
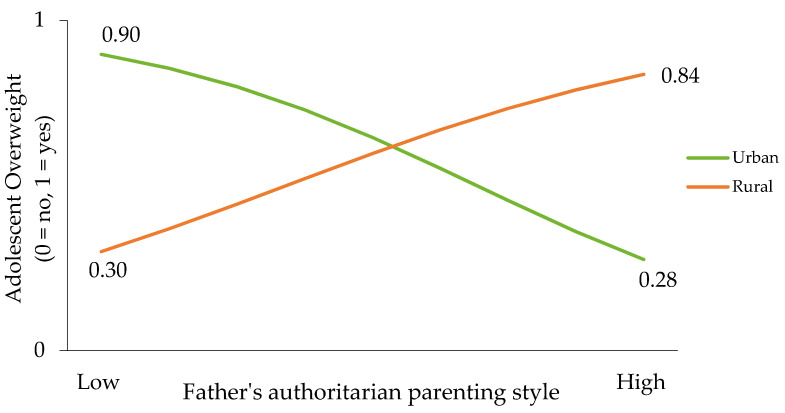
Moderating effect of area of residence on the relationship between authoritarian fathers and adolescent overweight.

**Figure 2 nutrients-14-05328-f002:**
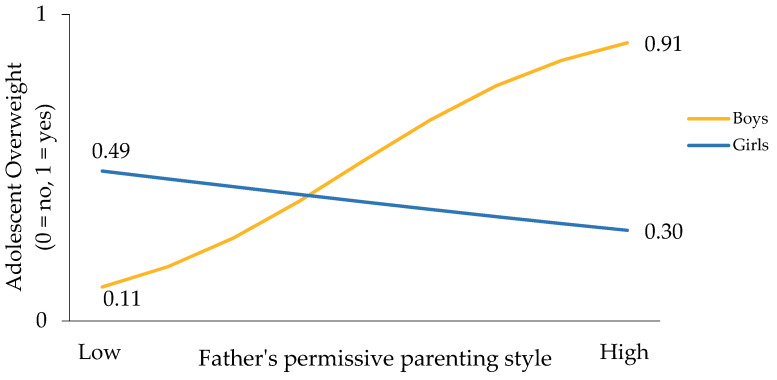
Moderating effect of sex on the relationship between permissive fathers and adolescent overweight in urban areas.

**Table 1 nutrients-14-05328-t001:** General and weight characteristics of the study population ^1,2^.

Study Population Characteristics	Overall (*n* = 695)		Sex				*p*-Value	Area of Residence				*p*-Value
			Girls *n* = 451 (65%)		Boys *n* = 244 (35%)			Urban *n* = 349 (50.2%)		Rural *n* = 346 (49.8%)		
Age (years)	14.9	(1.7)	14.9	(1.7)	14.9	(1.6)	0.874	14.9	(1.6)	15.0	(1.7)	0.314
BMI (kg/m^2^)	22.3	(4.3)	22.4	(4.7)	21.9	(4.0)	0.167	22.2	(4.3)	22.3	(4.6)	0.929
Healthy weight (%)	67.5	-	68.1	-	66.4	-	0.436	69.1	-	65.9	-	0.659
Overweight/obese (%) ^3^	32.5	-	31.9	-	33.6	-	0.532	30.9	-	32.5	-	0.079

^1^ Values are given as means (SD). ^2^ *p*-values were determined using independent sample *t*-tests for continuous measurements and chi-square for categorical variables. ^3^ Overweight includes adolescents categorized as overweight (BMI Z score for age ≥+1 and <+2) and obese (BMI Z score for age ≥+2).

**Table 2 nutrients-14-05328-t002:** Mean and standard deviation of fathers’ and mothers’ perceived parenting styles according to adolescent weight classification, and for the total sample.

Parenting Style	Fathers (*n* = 695)	SD	Mothers (*n* = 695)	SD
Healthy weight				
Authoritative	3.07	(1.06)	3.42	(0.97)
Permissive	2.18	(0.76)	2.36	(0.76)
Authoritarian	1.88	(0.61)	2.09	(0.64)
Overweight				
Authoritative	3.08	(1.06)	3.47	(0.94)
Permissive	2.20	(0.78)	2.39	(0.79)
Authoritarian	1.87	(0.56)	2.09	(0.66)
Total sample				
Authoritative	3.07 ^a^	(1.06)	3.43 ^a^	(0.96)
Permissive	2.19 ^b^	(0.77)	2.36 ^b^	(0.77)
Authoritarian	1.88 ^c^	(0.59)	2.08 ^c^	(0.65)

^a, b, c^ The means with different superscripts in each column differ significantly (*p* < 0.001) unless otherwise indicated.

**Table 3 nutrients-14-05328-t003:** Summary of hierarchical logistic analyses predicting the influence of fathers’ and mothers’ parenting styles on overweight among urban and rural Costa Rican adolescents (*n* = 695) ^1^.

Variable	*B* ^2^	Standard Error	Wald Chi-Square Test	*p*-Value ^3^	*Exp(B)* ^4^	95% CI for *Exp(B)*	
Age	−0.088	0.052	2.799	0.094	0.916	0.827	1.015
Sex (0 = Boys)	−0.104	0.186	0.312	0.576	0.901	0.626	1.298
Area (0 = Urban)	0.242	0.177	1.869	0.172	1.274	0.900	1.802
SES (0 = Low)	0.079	0.242	0.108	0.742	1.083	0.674	1.739
Authoritative Fathers	−0.254	0.336	0.571	0.445	0.776	0.401	1.499
Authoritarian Fathers	−0.159	0.374	0.181	0.671	0.853	0.410	1.775
Permissive Fathers	0.968	0.438	4.870	0.027	2.632	1.114	6.215
Authoritative Mothers	−0.239	0.385	0.384	0.536	0.788	0.37	1.676
Authoritarian Mothers	−0.039	0.376	0.011	0.917	0.962	0.461	2.008
Permissive Mothers	−0.398	0.417	0.914	0.339	0.672	0.297	1.519
Authoritative Fathers x Sex ^5^	0.762	0.419	3.306	0.069	2.143	0.942	4.875
Authoritarian Fathers x Sex	0.007	0.483	0.001	0.988	1.007	0.390	2.597
Permissive Fathers x Sex	−1.367	0.544	6.313	0.012	0.255	0.088	0.740
Authoritative Mothers x Sex	−0.013	0.466	0.001	0.978	0.988	0.396	2.463
Authoritarian Mothers x Sex	0.542	0.486	1.244	0.265	1.719	0.663	4.456
Permissive Mothers x Sex	0.343	0.523	0.431	0.512	1.410	0.506	3.930
Authoritative Fathers x Area	0.203	0.555	0.134	0.715	1.225	0.413	3.634
Authoritarian Fathers x Area	1.403	0.660	4.528	0.033	4.069	1.117	14.821
Permissive Fathers x Area	−1.677	0.685	5.988	0.014	0.187	0.049	0.716
Authoritative Mothers x Area	0.544	0.597	0.829	0.362	1.722	0.535	5.547
Authoritarian Mothers x Area	−0.839	0.619	1.837	0.175	0.432	0.128	1.454
Permissive Mothers x Area	1.131	0.618	3.342	0.068	3.098	0.922	10.41
Authoritative Fathers x Sex x Area	−0.601	0.657	0.835	0.361	0.548	0.151	1.989
Authoritarian Fathers x Sex x Area	−1.007	0.830	1.472	0.225	0.365	0.072	1.859
Permissive Fathers x Sex x Area	1.845	0.816	5.113	0.024	6.328	1.278	31.332
Authoritative Mothers x Sex x Area	−0.525	0.706	0.553	0.457	0.592	0.148	2.361
Authoritarian Mothers x Sex x Area	−0.213	0.793	0.072	0.788	0.808	0.171	3.822
Permissive Mothers x Sex x Area	−0.778	0.756	1.057	0.304	0.460	0.104	2.023

^1^ Parenting styles were centered on their means. Sex, area, and SES variables were dummy coded as appropriate. ^2^ *B* = coefficient for the constant in the model. ^3^ A *p*-value < 0.05 was considered statistically significant. ^4^
*Exp*(*B*) = exponentiation of B coefficient (Odds Ratio). ^5^ x = interaction between variables.

## Data Availability

The data presented in this study are available on request to the corresponding author.

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
