# Peer review of "Association between Fathers’ and Mothers’ Parenting Styles and the Risk of Overweight/Obesity among Adolescents in San José Province, Costa Rica"

_nutrients, 2022, doi:10.3390/nu14245328_

Round 1
Reviewer 1 Report
Dear Authors:
The authors have carried out the study “Association between Fathers’ and Mothers’ Parenting Styles and the Risk of Overweight/Obesity among Costa Rican Adolescents”. The objective of this study is to describe the association between fathers’ and mothers’parenting styles and adolescent overweight and obesity risk by the sex of the child and by their area of residence in Costa Rica.
The authors have conducted a comprehensive and scientifically correct study.
Nevertheles, some considerations need to be taken into account:
§ Fathers are often underrepresented in observational studies on parenting styles and obesity studies in general. What do you attribute it to? This fact should be further developed in the discussion.
§ Some other type of family structure has been analyzed in the study (single parent families, same-sex parents, e.g)?
§ Line 54: a should be changed for “are”
§ Table 2: What does a, b and c (superscript) stands for?
§ Line 300: substitue oher for “other”
§ Reference 30: substitue Sétimo for “Séptimo”
§ The study was limited to urban and rural areas within the province of San José, though the highest proportion of Costa Rican adolescents is clustered in that province. The sample was obtained from a very specific and limited geographical área. Therefore, the results are not nationally representative and should not be extrapolated at a national level. So, the tittle of the manuscript is tendentious and may lead to wrong interpretations.
§ References should be described as recommended by the style guide of the journal. Journals should be cited as Abbreviated Journal Name. Please review the reference list and follow the journal style guide (https://www.mdpi.com/journal/nutrients/instructions#preparation). Journal acronyms alternate with full titles. There is no homogeneity.
The study is well designed and is very attractive, but some very general conclusions are drawn from a very specific geographical area and extrapolated to an entire country, which could lead to an interpretation bias.
Kind regards
Author Response
RESPONSE TO REVIEWER 1
Dear reviewer,
Below you will find details of the changes we made to the manuscript in response to your recommendations and comments. To facilitate finding the changes, they were marked up using the 'Track Changes in the manuscript.
The authors have carried out the study “Association between Fathers’ and Mothers’ Parenting Styles and the Risk of Overweight/Obesity among Costa Rican Adolescents”. The objective of this study is to describe the association between fathers’ and mothers’ parenting styles and adolescent overweight and obesity risk by the sex of the child and by their area of residence in Costa Rica.
The authors have conducted a comprehensive and scientifically correct study.
Nevertheless, some considerations need to be taken into account:
- Fathers are often underrepresented in observational studies on parenting styles and obesity studies in general. What do you attribute it to? This fact should be further developed in the discussion.
A/ The topic was included in the discussion of the manuscript. Now it reads:
The underrepresentation of fathers in observational studies on parenting styles and the risk of overweight/obesity in pediatric population has been associated with the traditional roles of guardians based on gender, lack of time, lack of interest in research, and lack of availability and accessibility compared to mothers [64,65]. However, a recent study with more than 300 parents showed that nearly 80% did not participate in research because they were not invited [65]. Davison [65] has suggested that parents are more likely to participate in studies when they perceive that the time they must allocate is short, the recruitment material emphasizes the benefits their participation would have for them and their families, and they receive information about the study through a parent-sensitive approach.
Please see lines: 358-366
- Some other type of family structure has been analyzed in the study (single parent families, same-sex parents, e.g)?
A/ No, unfortunately no other variable related to family structure was collected in the study.
- Line 54: a should be changed for “are”
A/ Thank you. The grammatical correction has been made.
Please see line: 55
- Table 2: What does a, b and c (superscript) stands for?
A/ At the bottom of table 2 an explanation of the a, b, and c superscripts has been included. Now reads:
a,b,c The means with different superscripts in each column differ significantly (p<0.001) unless otherwise indicated.
Please see lines: 234-235
- Line 300: substitue oher for “other”
A/ Done. To improve the reading of the text, the word "other" was replaced by "Another"
Please see line 299
- Reference 30: substitute Sétimo for “Séptimo”
A/ Thank you, the correction was made.
Please see line 496
- The study was limited to urban and rural areas within the province of San José, though the highest proportion of Costa Rican adolescents is clustered in that province. The sample was obtained from a very specific and limited geographical area. Therefore, the results are not nationally representative and should not be extrapolated at a national level. So, the tittle of the manuscript is tendentious and may lead to wrong interpretations
A/ The title of the manuscript was modified. Now reads:
Association between Fathers’ and Mothers’ Parenting Styles and the Risk of Overweight/Obesity among Adolescents in San José Province, Costa Rica.
Please see lines 2-4
- References should be described as recommended by the style guide of the journal. Journals should be cited as Abbreviated Journal Name. Please review the reference list and follow the journal style guide
(https://www.mdpi.com/journal/nutrients/instructions#preparation). Journal acronyms alternate with full titles. There is no homogeneity.
A/ The format of the references has been corrected according to the style guide of the journal
- The study is well designed and is very attractive, but some very general conclusions are drawn from a very specific geographical area and extrapolated to an entire country, which could lead to an interpretation bias.
A/ The conclusions were focused only on the geographic area in which the study was conducted. Now reads:
In conclusion, our results suggest that fathers of adolescent inhabitants of the province of San José have the potential to influence their children’s weight outcomes, perhaps even over and above the mothers’ influence. This shows that fathers' and mothers' parenting styles are related differently to the risk of overweight/obesity in adolescents living in this province. Understanding the influence of both paternal and maternal parenting styles on adolescent eating behaviors will help build up the knowledge of the emotional climates in which positive and negative nutritional behaviors originate.
Please see lines: 367-373

Reviewer 2 Report
The authors well explored the association between parenting styles and the risk of overweight/obesity among Costa Rican Adolescents. However, I have several concerns about the study.
1. In the abstract, also in the data section, the authors must mention the period in which the case study was conducted.
2. The implications of this research should be emphasized.
3. Please, compute a measure of internal consistency. I recommend you to use it Cronbach's alpha?
Author Response
RESPONSE TO REVIEWER 2
Dear reviewer,
Below you will find details of the changes we made to the manuscript in response to your recommendations and comments. To facilitate finding the changes, they were marked up using the 'Track Changes in the manuscript.
The authors well explored the association between parenting styles and the risk of overweight/obesity among Costa Rican Adolescents. However, I have several concerns about the study.
- In the abstract, also in the data section, the authors must mention the period in which the case study was conducted.
A/ Thanks for the suggestion. Now reads:
Data are cross-sectional from a sample of adolescents (13-18 years old) enrolled in ten urban and eight rural schools (n = 18) in the province of San José, Costa Rica, in 2017.
Please see lines: 18-19
- The implications of this research should be emphasized.
A/ The implications of this research has been included in the discussion. Now reads:
Considering the relevant role of parents in the risk of overweight/obesity evidenced in this study, researchers should look for strategies and facilitate parental participation in the studies. We must expand our understanding of the critical public health implications of paternal parenting styles, particularly for designing and implementing culturally and gender-appropriate family interventions aimed at preventing and reducing overweight among adolescents living in the San José province.
Please see lines: 352-357
- Please, compute a measure of internal consistency. I recommend you use Cronbach's alpha?
A/ Internal consistency measures for the three subscales of the Parenting Styles and Dimensions Questionnaire were included in section 2.3. Parenting Styles Questionnaire. Now reads:
This validation has been published elsewhere [37], but in brief, the paternal and maternal authoritative and authoritarian parenting styles scales had an internal consistency that ranged between satisfactory and good [38] (paternal: Cronbach α = 0.91 and 0.70, respectively; maternal: Cronbach α = 0.90 and 0.73, respectively). However, the permissive parenting style scale had a moderate consistency for mothers (Cronbach α = 0.51) and fathers (Cronbach α = 0.52).
Please see lines: 153-158
